# Strong coupling between a microwave photon and a singlet-triplet qubit

J. H. Ungerer [1,2,7] ✉, A. Pally [1,7] ✉, A. Kononov [1], S. Lehmann [3], J. Ridderbos [1,6], P. P. Potts [1,2], C. Thelander [3], K. A. Dick [4], V. F. Maisi [3], P. Scarlino [5], A. Baumgartner [1,2] & C. Schönenberger [1,2]

Combining superconducting resonators and quantum dots has triggered tremendous progress in quantum information, however, attempts at coupling a resonator to even charge parity spin qubits have resulted only in weak spin-photon coupling. Here, we integrate a zincblende InAs nanowire double quantum dot with strong spin-orbit interaction in a magnetic-field resilient, high-quality resonator. The quantum confinement in the nanowire is achieved using deterministically grown wurtzite tunnel barriers. Our experiments on even charge parity states and at large magnetic fields, allow us to identify the relevant spin states and to measure the spin decoherence rates and spin-photon coupling strengths. We find an anti-crossing between the resonator mode in the single photon limit and a singlet-triplet qubit with a spin-photon coupling strength of $g/2\pi = 139 \pm 4$ MHz. This coherent coupling exceeds the resonator decay rate $\kappa/2\pi = 19.8 \pm 0.2$ MHz and the qubit dephasing rate $\gamma/2\pi = 116 \pm 7$ MHz, putting our system in the strong coupling regime.

Spin qubits in semiconductors are promising candidates for scalable quantum information processing due to long coherence times and fast manipulation[1–4]. For the qubit readout, circuit quantum electrodynamics based on superconducting resonators[5], allows a direct and fast measurement of qubit states and their dynamics[6]. Recently, resonators were used to achieve charge–photon[7,8], spin–photon[9–11] as well as coherent coupling of distant charge[12] and spin qubits[13,14], enabling coherent information exchange between distant qubits. However, the small electric and magnetic moments of individual electrons require complicated device architectures such as micromagnets, and a large number of surface gates that render scaling up to more complex architectures challenging. These approaches typically achieve a relatively weak electron spin–photon coupling on the order of ~10–30 MHz. In addition to single electron spin qubits, also spin qubits based on two electrons in a double quantum dot (DQD), e.g. a singlet–triplet qubit have been demonstrated[15]. Spin qubits based on two electrons typically offer a large hybridization of the spin and charge degree of freedom compared to single-electron spin qubits in principle allowing even stronger coupling strengths. So far, however, the experimentally achieved coupling strengths in such systems[16,17] remained well below the strong coupling limit in which the coherent coupling rate exceeds both the cavity mode decay rate and the qubit linewidth.

Here, we demonstrate that the strong coupling regime between a singlet–triplet qubit and a single photon in a superconducting resonator can be reached. We achieve this strong coupling by carefully designing the resonator and by using a DQD defined by in-situ grown tunnel barriers in a semiconductor with a large spin–orbit interaction. The tunnel barriers consist of InAs segments in the wurtzite crystal-phase with an atomically sharp interface to the zincblende bulk of the nanowire (NW)[18]. These crystal-phase barriers are highly reproducible and render the need of barrier gates obsolete, simplifying integration with superconducting resonators and making the nanowires a viable prototype for scalable quantum computing architectures.

[1]Department of Physics, University of Basel, Klingelbergstrasse 82, CH-4056 Basel, Switzerland. [2]Swiss Nanoscience Institute, University of Basel, Klingelbergstrasse 82, CH-4056 Basel, Switzerland. [3]Solid State Physics and NanoLund, Lund University, Box 118, S-22100 Lund, Sweden. [4]Centre for Analysis and Synthesis, Lund University, Box 124, S-22100 Lund, Sweden. [5]Institute of Physics and Center for Quantum Science and Engineering, Ecole Polytechnique Fédérale de Lausanne, CH-1015 Lausanne, Switzerland. [6]Present address: MESA+ Institute for Nanotechnology, University of Twente, P.O. Box 217, 7500 AE Enschede, The Netherlands. [7]These authors contributed equally: J. H. Ungerer, A. Pally. ✉e-mail: jungerer@g.harvard.edu; alessia.pally@unibas.ch

In this work, we make use of the large spin–orbit interaction in these nanowires[19] to define a singlet–triplet qubit at a finite in-plane magnetic field in which the $T_{1,1}^+$ and $S_{2,0}$ states hybridize, forming a quantum two-level system. Incorporating a NW with a magnetic-field resilient resonator based on NbTiN[20,21] allows us to measure an avoided crossing between the singlet–triplet qubit and a single-photon excitation of the resonator at a magnetic-field strength of $B = 300$ mT. The measured coupling strength is very large compared to previously reported electron spin–photon coupling[9–11], which enables us to reach the strong coupling regime. In addition, by analyzing the response of the hybridized resonator-qubit system for varying magnetic-field strengths, we perform qubit spectroscopy[22–24]. This allows us to identify the specific spin states and to quantitatively extract the relevant device properties.

## Results

### Device characterization

Details about the NW properties and their growth can be found in the supplementary. The resonator-qubit system of device A is shown in Fig. 1a, including a false-colored SEM-image of the crystal-phase defined NW DQD. We report similar experiments for two devices, A and B, with B discussed in the supplementary. They are measured in a dilution refrigerator with a base temperature of 70 mK. The DQD forms in the 490 and 370 nm long zincblende segments (green), separated by 30 nm long wurtzite (red) tunnel barriers with a conduction band offset of -100 meV[25], as illustrated in Fig. 1b. A high-impedance, half-wave coplanar-waveguide resonator is capacitively coupled to the DQD at its voltage anti-node via a sidegate. In addition, the same sidegate can be used to tune the DQD charge states using a dc voltage ($V_R$) applied at the resonator voltage node. The DQD state is probed by reading out the resonator rf-transmission. We extract the bare resonance frequency of the resonator $\omega_0/2\pi = 5.1705 \pm 0.0003$ GHz at zero magnetic field and the bare decay

rate $\kappa|_{B=0}/2\pi = 27.3 \pm 0.6$ MHz. The resonator design and fitting are described in detail in the "Methods" section. In the following, we prepare the DQD in an even charge configuration in the many-electron regime (see the "Methods" section), described by a two-electron Hamiltonian given in the "Methods" section. Figure 1c shows the eigenvalues of this Hamiltonian as a function of external magnetic field $B$ at a fixed DQD detuning. At zero magnetic field, the detuning renders the singlet $S_{2,0}$ the ground state, for which both electrons reside in the same dot. Without spin-rotating tunneling, this, and the $S_{1,1}$ state, with the electrons distributed to different dots, form a charge qubit[26]. The subscripts describe the dot electron occupation of the left and right dots, respectively. By applying an external magnetic field, the Zeeman effect lowers the energy of the triplet $T_{1,1}^+$ state, that becomes the ground state for sufficiently high magnetic fields. In the presence of a spin-rotating tunneling $t = \Delta_{SO}/2$ induced by the intrinsic spin–orbit interaction $\Delta_{SO}$, the energy levels of the hybridized $S_{2,0}$ and $T_{1,1}^+$ states are split. The two new eigenstates of the avoided crossing form a singlet–triplet qubit shown schematically in Fig. 1a and b.

Figure 2a shows the charge stability diagram of device A at a magnetic field of 600 mT with the angle $\alpha = 164°$ with respect to the NW axis (see Fig. 1a) detected as a shift in the transmission phase $\varphi$ of the resonator, plotted as a function of the two gate voltages $V_L$ and $V_R$ at a fixed probe frequency of $\omega_p/2\pi = 5.174$ GHz, close to resonance. We observe a characteristic honeycomb pattern of the charge stability diagram of a DQD. Using a capacitance model[27,28], we extract the gate-to-dot capacitances $C_{R2} = 44 \pm 2$ aF, $C_{L2} = 2.0 \pm 0.2$ aF, $C_{R1} = 5 \pm 2$ aF and $C_{L1} = 4.6 \pm 0.2$ aF for device A.

We now focus on one particular inter-dot transition (IDT) marked by a green rectangle in Fig. 2a. The same IDT is shown in Fig. 2b and c at $B = 0$ T and $B = 300$ mT respectively, with $\alpha = 57°$. In Fig. 2d we show the normalized transmission $(A/A_0)^2$ at $B = 0$ T, while varying the probe frequency $\omega_p$ and relative detuning $\varepsilon_{rel}$, illustrated by the white line in Fig. 2b. An electron can now reside on either of the two tunnel-coupled dots, constituting a charge qubit. At the IDT, close to charge degeneracy, the electrical dipole moment of the charge qubit interacts with the resonator, resulting in a dispersive shift of the resonance frequency. By fitting input-output theory (see the "Methods" section) to this particular IDT, we extract the inter-dot tunnel coupling $t|_{B=0}/2\pi = 5.1 \pm 1.0$ GHz, the charge–photon coupling $g_0|_{B=0}/2\pi = 353 \pm 72$ MHz, and the charge qubit linewidth $\gamma|_{B=0}/2\pi = 1.7 \pm 0.7$ GHz.

### Strong spin–photon coupling

When investigating the magnetic-field dependence of IDTs similar to the ones shown in Fig. 2b, c, we observe two qualitatively different behaviors which we identify as even and odd charge parity configurations described in the "Methods" section. In the following, we investigate a single IDT, shown in Fig. 2c, with an even charge parity.

As illustrated in Fig. 1c, the DQD can be operated as a singlet–triplet qubit when applying a magnetic field. The qubit frequency $\omega_q$ can be brought into resonance with the cavity frequency $\omega_0$ at $B \gtrsim 200$ mT, as discussed in more detail below. At the resonance condition $(\omega_q \sim \omega_0)$, an anti-symmetric (bonding) and a symmetric (anti-bonding) qubit-photon superposition state are formed. The corresponding resonances can spectroscopically be discriminated only if the splitting $2g$ between them is larger than the dressed states' linewidth $\gamma + \kappa/2$[29]. In particular, the hybrid system is considered strongly coupled if the qubit-photon coupling strength $g$ exceeds $\gamma$ and $\kappa$[29].

In Fig. 3a, we plot a spectroscopic measurement of the resonator where the singlet–triplet qubit is tuned into resonance by applying an electrostatic detuning $\varepsilon_{rel}$ relative to the configuration at which $S_{2,0}$ and $T_{1,1}^+$ would be fully degenerate in the absence of a a spin-rotating tunneling. Consistent with strong coupling, we observe an avoided crossing between the resonator and the qubit. At the points where the bare qubit frequency $\omega_q$ and resonator frequency $\omega_0$ (dashed, white

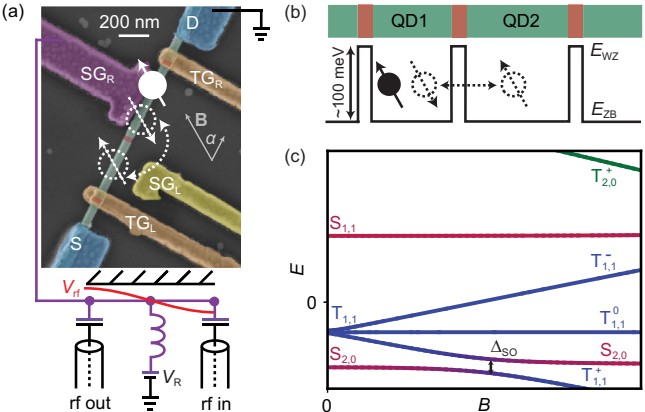

**Fig. 1 | Coupled resonator-qubit system. a** False colored SEM-image of device A. The NW (green) is divided into two segments by an in-situ grown tunnel barrier (red), thus forming the DQD system. The NW ends are contacted by two Ti/Au contacts (S,D) and the NW segments can be electrically tuned by two Ti/Au sidegates $SG_R$ (purple) and $SG_L$ (yellow). The voltage anti-node with amplitude $V_{rf}$ of a high-impedance, half-wave resonator is connected to $SG_R$. Top gates $TG_L$ and $TG_R$ (orange) are kept at a constant voltage of $-0.28$ V. The magnetic field is applied in-plane at an angle $\alpha$ with respect to the NW axis, as illustrated by the gray arrow. The white arrows illustrate an even charge configuration with the two degenerate DQD states $T_{1,1}^+$ and $S_{2,0}$. **b** Schematic of the crystal-phase defined DQD. The conduction band of wurtzite at energy $E_{wz}$ and the one of zincblende at energy $E_{zb}$ are offset by -100 meV, resulting in a tunnel barrier between the zincblende segments. The intrinsic spin–orbit interaction enables spin-rotating tunneling between these segments. **c** Energy levels of an even charge configuration as a function of magnetic field $B$ at a fixed positive detuning $\varepsilon$ between the dot levels. At finite magnetic fields, $T_{1,1}^+$ (blue) hybridizes with $S_{2,0}$ (red) defining a singlet-triplet qubit with an energy splitting given by the spin–orbit interaction strength $\Delta_{SO}$.

curves) are degenerate, instead of crossing, they anti-cross. And in Fig. 3a, a faint double peak structure is visible at around $\varepsilon_{rel} \sim 0$ as $2g > \kappa/2 + \gamma$, signature of the strong coupling regime[29].

For a quantitative analysis, we fit Lorentzians to the transmission of each trace of constant $\varepsilon_{rel}$, we extract the transition frequencies $\omega_\pm$ of the dressed states. These are fitted to the Jaynes–Cummings model (solid, white curves in Fig. 3a) described in the "Methods" section. From this fit, we extract the tunnel rate $t|_{B=300mT}/2\pi = \Delta_{so}|_{B=300mT}/4\pi = 2.54 \pm 0.03$ GHz and bare spin–photon coupling strength $g_0^{JC}|_{B=300mT}/2\pi = 123 \pm 16$ MHz. The extracted tunnel rate allows to plot the qubit transition frequency $\omega_q = \sqrt{(\Delta_{so}/\hbar)^2 + (\varepsilon_{rel})^2}$ in Fig. 3a and to identify the resonance condition $\omega_q = \omega_0$ at a small electrostatic detuning $\varepsilon_{rel}/2\pi = \pm 1.0$ GHz. We evaluate the effective coupling strength $g = g_0 \cdot 2t/\omega_q$ at the finite detuning $\varepsilon_{rel}/2\pi = -1.0$ GHz and obtain $g^{JC}|_{\varepsilon_{rel}=-1GHz}/2\pi = 121 \pm 16$ MHz, as the spin–photon coupling strength on resonance condition.

In Fig. 3b, we plot a line trace at this detuning value as indicated in Fig. 3a. Despite the large noise, the double peak structure is also clearly visible and stands in stark contrast to the bare resonator transmission at large detuning (see corresponding linetrace in Fig. 3c). Using Eq. (6) derived from input–output theory described in the supplementary, we fit these data at 300 mT and extract the spin-photon coupling strength $g_{\varepsilon_{rel}=-1GHz}/2\pi = 139 \pm 4$ MHz and qubit dephasing $\gamma/2\pi = 116 \pm 7$ MHz where we used the bare resonator decay $\kappa|_{B=300mT}/2\pi = 19.8 \pm 0.6$ MHz. This value agrees well with the one obtained from the Jaynes–Cummings model. Using the values from input–output theory we model the whole anti-crossing using input–output theory in Fig. 3d, observing a very good agreement with the measurement.

All together, this measurement therefore clearly demonstrates that the strong coupling regime between a single microwave photon and a singlet–triplet qubit is reached.

## Magnetospectroscopy

To explicitly identify and characterize the spin–orbit eigenstates and to independently verify the character of the singlet–triplet qubit, we now study the magnetic field evolution of the IDT from 0 up to 900 mT applied at the angle $\alpha = 130°$. We measure the amplitude $A$ and phase $\varphi$ of the signal transmitted through the resonator as function of detuning $\varepsilon$ and magnetic field strength $B$. A non-zero $\varphi$ occurs at the IDT when tunneling between the dots is allowed resulting in a non-zero DQD dipole moment. As described in the "Methods" section, we model the DQD by an effective two electron Hamiltonian which allows us to fit the gate voltage and field dependence of the IDT (white dashed line in Fig. 4a). We find that the magneto-dispersion of the IDT is well described using the following fit parameters: the spin-conserving singlet and triplet tunnel rates $t_c^S/2\pi \approx 8.5$ GHz, and $t_c^T/2\pi \approx 3.2$ GHz, the singlet–triplet coupling rate $t_{SO}/2\pi = \Delta_{SO}/4\pi \approx 2.9$ GHz, the electron $g$-factors of the right and left dots, $g_R \approx 1$ and $g_L \approx 8$, as well as the single dot singlet–triplet energy splitting $\Delta_{ST}/2\pi \approx 47$ GHz. These fit parameters are consistent with parameters obtained previously in this material system[19,30–33]. We note, however, that the fit is under-determined and therefore, it does not provide accurate numbers. Nonetheless, the model gives a qualitative, physical understanding of the system and allows us to establish which DQD levels interact with the resonator.

Independently, we gain quantitative information about the system by considering the functional dependence of the amplitude $A$ and

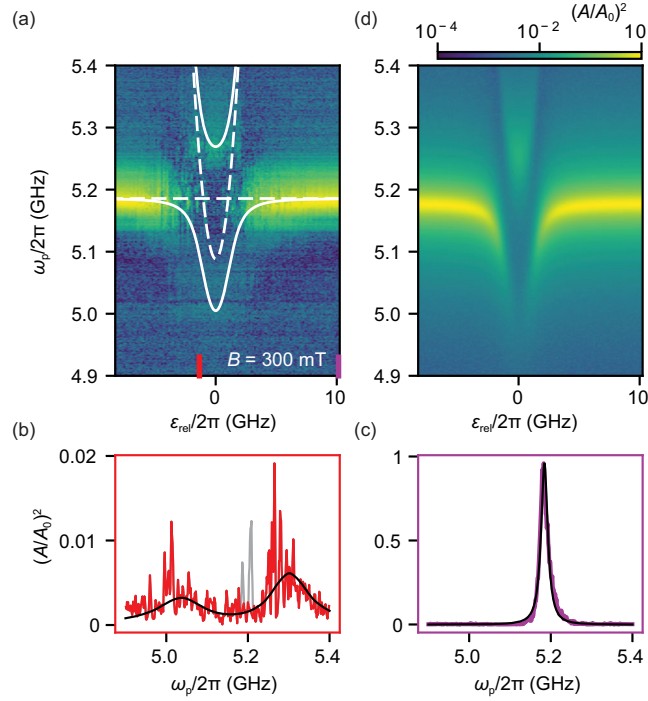

**Fig. 3 | Strong spin–photon coupling. a** Anti-crossing of the resonator and the qubit found when plotting the resonator transmission as a function of detuning $\varepsilon_{rel}$ and probe frequency $\omega_p$ at a magnetic field of $B = 300$ mT and $\alpha = 57°$. The solid white curves are the eigenstate energies from fits to a Jaynes–Cummings model (Eq. (2) in the "Methods" section. The faint double-peak structure at $\varepsilon \approx 0$ is an unambiguous signature of the strong coupling regime, $g > \kappa, \gamma$[29]. **b, c** Cross sections at the detunings indicated by colored bars in (**a**). The solid lines stem from a fit to input–output theory. **b** Double-peak structure at $\omega_q \sim \omega_0$ (see text). The larger noise floor for $\omega_p \sim \omega_0$ (gray data) is attributed to the bare resonator which is visible in spectroscopy because of a finite coupling between DQD and leads resulting in an odd DQD occupation for a short fraction of time during data acquisition. **c** Transmission for $\omega_q \gg \omega_0$, corresponding to the bare resonator. **d** Simulation using input–output theory with the parameters extracted from the input–output fit to (**b**). For these measurements, given the input-power $P_{in} = -133$ dBm, the average number of photons is $n < 0.25$ (see the "Methods" section).

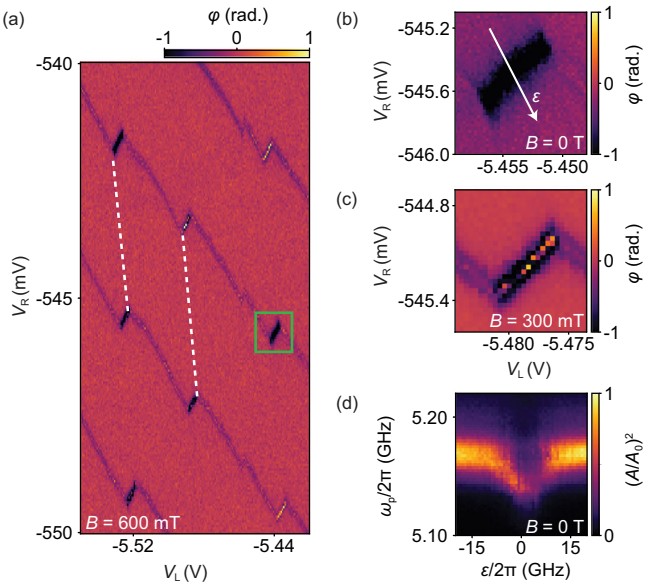

**Fig. 2 | Dispersive sensing of the DQD at $B = 0$. a** Charge stability diagram of the device at $B = 600$ mT applied at $\alpha = 164°$ with respect to the NW, in which the resonator phase $\varphi$ is measured as a function of the SG voltages $V_R$ and $V_L$. A zoom on the interdot transition pointed out by the green rectangle is shown in **b** and **c** at $B = 0$ T and $B = 300$ mT with $\alpha = 57°$, respectively. **d** Resonator transmission $(A/A_0)^2$ versus probe frequency $\omega_p$ and detuning $\varepsilon$ (illustrated by the white line in **b**). At the charge degeneracy point of the DQD, we find a dispersive shift of $21 \pm 2$ MHz with respect to the bare resonance frequency. At small positive detuning a triplet state crosses the IDT, leading to a suppressed resonator transmission.

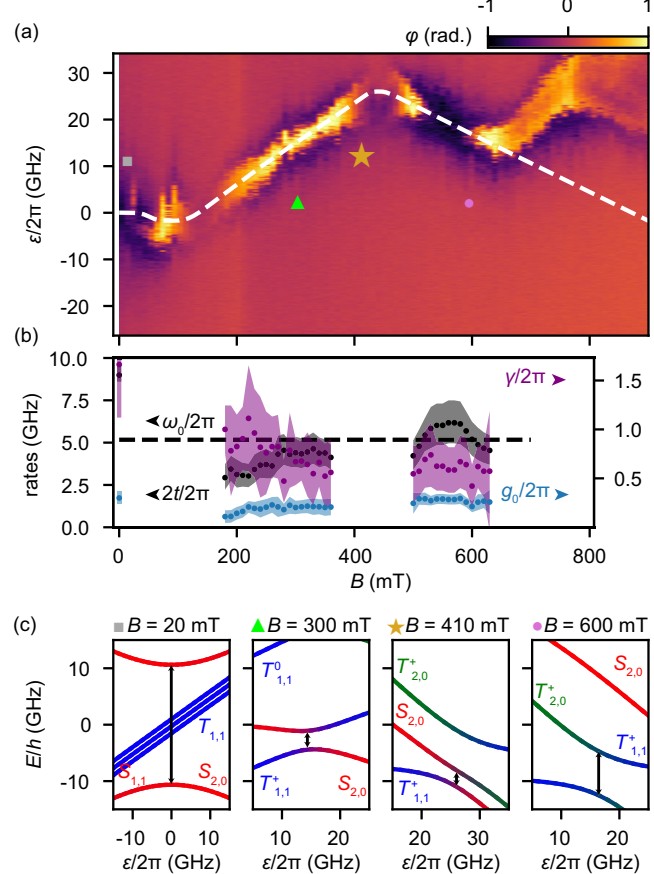

**Fig. 4 | Magnetospectroscopy of the singlet-triplet qubit. a** Dispersive shift $\chi$ as a function of the magnetic field $B$ at an angle of $\alpha = 130°$ and detuning $\varepsilon$. The white dashed line is a fit of the effective two-electron Hamiltonian (Eq. (12)) to the data. **b** Extracted tunnel rate $2t/2\pi$ (black), qubit-photon coupling $g_0/2\pi$ (blue) and qubit linewidth $\gamma/2\pi$ (purple). The bare resonator frequency is indicated by the dashed black line. Shaded areas indicate the errorbars which originate from the uncertainty of the gate lever arm, which was independently measured. **c** Two-electron energy level diagrams at various magnetic fields with the corresponding field strength indicated in **a** and **b** by the given symbols. For clarity a constant offset of 10, 20, and 30 GHz was added to the energy levels at 300, 410 and 600 mT. Given the input power $P_{in} = -128$ dBm, the average photon number is $n < 0.8$ in these experiments (see methods).

phase $\varphi$. This is possible because the resonator provides an absolute energy scale allowing for a quantitative analysis of the interaction between the DQD and the resonator and hence to perform qubit spectroscopy[22-24]. This spectroscopy complements the preceding DQD Hamiltonian fit. As described in the "Methods" section, by fitting input–output theory to $\varphi$ and $A$ simultaneously, we extract the qubit tunnel amplitude $t$, the qubit linewidth $\gamma$, and the qubit–photon coupling strength $g$ as a function of $B$, which we plot in Fig. 4b. Here, we assume $\gamma$ as constant in detuning $\varepsilon$.

Using the fits to both, the 2-electron Hamiltonian model and input–output theory in the 2-level approximation, allows us to directly identify several regimes, in each of which the qubit has a different spin-character. Figure 4c shows the corresponding DQD level structure based on the fit parameters as a function of $\varepsilon$ for different magnetic field.

At a low magnetic fields around $B = 20$ mT, the triplet states (blue curves) are Zeeman split and the ground-state curvature is dominated by the anti-crossing between $S_{1,1}$ and $S_{2,0}$ (red curves). We find a singlet charge qubit in the weak coupling limit, i.e. the linewidth exceeds the charge–photon coupling by a factor of five. The formation of an asymmetric double-dip structure in $\varphi(\varepsilon)$

between $B \sim 0.01$ T and $B \sim 0.18$ T is explained by an interaction between the three states $S_{2,0}$, $S_{1,1}$ and $T_{1,1}^+$ as described in the supplementary material. Traces of $\varphi(\varepsilon)$ with an asymmetric double-dip structure cannot be described by a two-level input–output model and are therefore not analyzed quantitatively here. At $B \approx 50$ mT, $\varphi$ becomes positive. Which we interpreted as a drop of the tunnel rate below the resonator frequency, $2t < \omega_0$.

As $B$ is increased, the triplet state $T_{1,1}^+$ becomes the ground state for $\varepsilon < 0$, as shown in the second panel of Fig. 4c for $B = 300$ mT. The spin–orbit interaction couples the singlet and triplet states, leading to an anti-crossing between $S_{2,0}$ and $T_{1,1}^+$, which constitutes a singlet–triplet qubit with $\omega_q = \Delta_{SO} = 2t_{SO}$[34,35]. In this regime, at larger $B$, the resonance condition between $S_{2,0}$ and $T_{1,1}^+$ occurs at larger $\varepsilon$, because the energy of the bare $T_{1,1}^+$ state decreases with larger $B$ and the energy of $S_{2,0}$ decreases with larger $\varepsilon$. Therefore, the IDT is observed at larger $\varepsilon$ for increasing $B$.

Consistent with the interpretation of the formation of a singlet–triplet qubit, we measure an approximately constant tunneling rate $t$ between $B \sim 0.18$ T and $B \sim 0.36$ T. In this regime, we extract the average spin–orbit tunneling rate to be $\bar{t}_{so} = 1.94 \pm 0.02$ GHz. At a magnetic field of $B \approx 370$ mT, the resonator phase $\varphi$ starts to vanish due to the the triplet state $T_{2,0}^+$ becoming relevant. The triplet state results in a level repulsion between $T_{2,0}^+$ and $T_{1,1}^+$ and hence leads to a reduced energy gap between the $S_{2,0}$ level and the $T_{1,1}^+$ level. In Fig. 4c, this is illustrated by the smaller energy gap (black arrow) at $B = 410$ mT compared to the one at $B = 300$ mT. Due to the reduced gap ($\Delta_{SO} << \omega_0$), the resonator-qubit coupling is reduced and hence is the singal in $\varphi$.

The level structure at large magnetic fields is plotted exemplary for $B = 600$ mT in the right panel of Fig. 4c. In this regime, the ground-state of the DQD at the IDT is formed by a superposition of the $T_{2,0}^+$ and $T_{1,1}^+$ states. We find that the curve of Fig. 4a turns back towards lower $\varepsilon$ for increasing $B$, which can be understood by noting that the spin-polarized triplets $T_{2,0}^+$ and $T_{1,1}^+$ form a charge qubit similar to the singlets at low field. While the transition is increasingly dominated by the triplet-charge qubit for increasing $B$, $\varphi$ becomes negative at the IDT, because the anti-crossing between the triplet states $T_{2,0}^+$ and $T_{1,1}^+$ occurs at much larger frequencies, $2t_c^T > 2t_{SO}, \omega_0$. Hence, the triplet charge qubit frequency does not cross the resonator frequency, leading to a negative phase shift.

At fields $B > 700$ mT the dispersion turns to higher $\varepsilon$ again. Which is not accounted for in our model. A possible explanations to this discrepancy is that the magnetic field not only affects the detuning $\varepsilon$ of the DQD but also the total energy. This results in the lead to dot transitions starting to influece the IDT at high magnetic fields. Nevertheless, the data is well described at the magnetic field strengths we investigate in detail.

This large number of detailed findings justify the parameters of the two-electron Hamiltonian introduced above, which, in turn, directly allows us to identify the singlet–triplet spin qubit, for which we find the strong coupling limit to the electromagnetic cavity.

Note, that the extracted qubit linewidth is larger in Fig. 4b compared to the strong-coupling in Fig. 3. This is caused by applying the magnetic field at different angles in the two measurements. The dependence of the qubit parameters on the angle of the in-plane magnetic field is beyond the scope of this manuscript and will be investigated in future studies.

## Discussion

In summary, we demonstrate a semiconductor nanowire DQD device with crystal-phase defined tunnel barriers that can be operated as different types of qubits, coupled to a high-impedance, high magnetic field resilient electromagnetic resonator. As the main result, we find a

singlet–triplet qubit for which we extract the relevant qubit parameters, especially a large electron spin–photon coupling of $g/2\pi = 139$ MHz in the single photon limit, reaching the strong coupling regime $g > \gamma, \kappa$.

Our experiments demonstrate that deterministically grown tunnel barriers allow for a reduced number of gate lines, and that, mediated by intrinsic spin–orbit interaction, singlet–triplet qubits can reach the strong coupling limit for low photon numbers, similar to flopping mode spin qubits[36,37]. This finding is potentially applicable to other promising platforms with strong spin–orbit interactions, like holes in Ge[35]. Our nanowire platform without depletion gates results in a significantly reduced gate-induced photon-leakage in the absence of on-chip filtering[6,38,39]. And, since DQD parameters (such as charging energy and individual tunnel rates) can be set deterministically in the NW growth, multiple NWs with optimal and essentially identical characteristics properties can be obtained simultaneously[40] and possibly integrated on the same substrate[41]. This drastically simplifies the search for an optimal gate regime and renders further gates, such as the top gates in our device, unnecessary. An optimized gate design with resonators with larger impedance[28] therefore presents an ideal platform to investigate new phenomena in the ultrastrong coupling regime[28,42]. Additionally, the large electron spin–photon coupling found in our experiments will be crucial for the implementation of two-qubit gates between distant spin qubits, a milestone on the way towards scalable quantum computers.

## Methods
### Resonator characterization and analysis
The resonator is fabricated from a thin-film NbTiN (thickness ~ 10 nm), sputtered onto a Si/SiO$_2$ (500 μm/100 nm) substrate[21]. These resonators can be operated for in-plane fields exceeding 5 T[20,21]. The large sheet kinetic inductance of the used NbTiN film of $L_{sq} \approx 90$ pH combined with the narrow center conductor width of ~380 nm, and the large distance to the ground plane of ~35 μm results in an impedance of 2.1 kΩ. The resonator can be dc biased using a bias line which contains a meandered inductor ensuring sufficient frequency detuning between the half-wave resonance used in the experiment and a second, low-quality resonance mode at a lower frequency that forms due to the finite inductance of the bias line[39]. A scanning electron micrograph of the resonator center-conductor is shown in Fig. S1b in the supplementary. One of the two resonator voltage anti-nodes is galvanically connected to gate SG$_R$ shown in Fig. 1c of the main text.

### Device fabrication
For the NW growth refer to Supplmentary II. After the resonator fabrication, the NWs are deposited on the device area using a micromanipulator. Following the deposition, the NW position and barrier locations are determined using scanning electron imaging. A GaSb-shell, grown for the barrier determination, is then removed before contacting by a 3 min 30 s wet-etching process using MF-319 developer[43,44]. The contacts and gates are then fabricated using standard e-beam lithography and e-beam evaporation. For the contacts, the native oxide is removed using in-situ argon-milling in the evaporation chamber. For the SGs and TGs no argon-milling is performed and the native oxide is left intact to serve as an insulating layer for the TGs.

### Charge parity determination
We measure the phase $\varphi$ and amplitude $A$ of the resonator as a function of detuning $\varepsilon$ and magnetic field $B$ at a probe-frequency $\omega_p/2\pi = 5.253$ GHz, close to the bare resonator frequency. A change in $\varphi$ reflects the dispersive interaction between the resonator and two anticrossing levels of the DQD[45,46]. Therefore,

the non-zero phase response of the resonator tracks the position of the IDT along the detuning axis. The comparison of the magnetic field dependence of the IDT position to a Hamiltonian model of the DQD allows one to determine the charge parity[46,47]. Figure S2a and S2b in the Supplementary show two typical low field IDT characteristics of device B.

For an odd number of electrons (Fig. S2b in the Supplementary), the DQD resonance gate voltage $V_R$, at which the IDT is observed, disperses linearly with magnetic field starting from zero. This can be understood considering the Zeeman-splitting of the unpaired electron energy levels and two non-equal Landé g-factors of the two dots. Figure S2c in the Supplementary shows the energy level diagram of a one-electron Hamiltonian including Zeeman-splitting with a g-factor difference of 1.0 and spin–orbit interaction $t_{SO}/2\pi = 5$ GHz at a magnetic field of $B = 500$ mT (green, dashed line in Fig. S2b. The one-electron Hamiltonian is explicitly discussed in the Supplementary material. The arrow points out the center of the IDT (largest curvature of the groundstate[48]) which corresponds to the largest dipole moment of the DQD and thus to the largest change in $\varphi$. This point shifts with $B$ towards increasingly negative values.

For an even number of electrons in the DQD at zero magnetic field (Fig. S2a in the Supplementary), a single dip in phase is observed, but at a low magnetic fields, $B \approx 60$ mT, a double dip structure emerges as a function of $\varepsilon$ (see Supplementary material for details). This double-dip originates from an interaction between $S_{2,0}$, $S_{1,1}$ and $T_{1,1}^+$ as explained in detail in the supplementary material. The dependence of the IDT on magnetic field for an even number of electrons can be understood using an effective two electron Hamiltonian including spin–orbit interaction described in more detail below. In Fig. S2c in the Supplementary, we plot the energy levels at a magnetic field $B = 0.15$ T. In contrast to the odd filling, starting at zero magnetic field, the arrow marking the center of the IDT barely changes, consistent with our measurement. The double dip vanishes when further increasing the magnetic field, because of an increasing occupation of the polarized triplet states. Once the Zeeman energy of the triplet state $|T_{1,1}^+\rangle$ becomes comparable to the singlet charge tunneling $t_c^S$, the position of the IDT as a function of $B$ disperses towards larger $\varepsilon$[47,49,50]. This transition is marked by the white dashed line at 0.2 T in Fig. S2a in the Supplementary.

Based on the good qualitative agreement between our data and the one electron and two electron Hamiltonian, respectively, we can clearly identify the even and odd charge parities.

### Jaynes–Cummings model
In the regime of only two DQD levels being relevant, we model the DQD Hamiltonian as an effective two-level system (qubit) interacting with a single mode in the resonator. The combined system is described by the Jaynes–Cummings model[51]

$$\hat{H}/\hbar = \omega_0 \hat{a}^\dagger \hat{a} + \frac{\omega_q}{2}\hat{\sigma}_z + g\left(\hat{a}\hat{\sigma}^\dagger + \hat{a}^\dagger \hat{\sigma}\right), \tag{1}$$

where $\hat{a}$ is the photon annihilation operator, $\hat{\sigma}$ the qubit lowering operator, and $\hat{\sigma}_z$ the Pauli z-matrix in the qubit subspace. The qubit frequency is given by $\omega_q = \sqrt{(2t)^2 + \varepsilon^2}$[26] with the effective qubit–photon coupling strength $g = g_0 \cdot 2t/\omega_q$ accounting for the mixing angle[7,52], where $g_0$ is the bare qubit–photon coupling. An excitation from the ground state has the transition frequency[52]

$$\omega_\pm = \frac{\omega_0 + \omega_q}{2} \pm \frac{1}{2}\sqrt{4g^2 + (\omega_0 - \omega_q)^2}. \tag{2}$$

## Input–output theory

To derive the response of the resonator, we use the equations of motion[53]

$$\partial_t \langle \hat{a} \rangle(t) = -i\omega_0 \hat{a}(t) - ig\langle \hat{\sigma} \rangle(t) - \frac{\kappa}{2}\langle \hat{a} \rangle(t)$$
$$-\sqrt{\kappa_1}\langle \hat{b}_{\text{in},1} \rangle(t) - \sqrt{\kappa_2}\langle \hat{b}_{\text{in},2} \rangle(t),$$
$$\partial_t \langle \hat{\sigma} \rangle(t) = -i\omega_q \langle \hat{\sigma} \rangle(t) + ig\langle \hat{a}\hat{\sigma}_z \rangle(t) - \gamma \langle \hat{\sigma} \rangle(t). \tag{3}$$

The input couplings are denoted by $\kappa_j$ and the operators $\hat{b}_{\text{in},j}(t)$ capture a coherent drive in port $j$. In our experiments $\kappa_1 \approx \kappa_2 \approx \kappa/2$ as the resonator is symmetrically coupled and operates in the strongly over-coupled regime. The output of the cavity can be computed from the input–output relation[53]

$$\langle \hat{b}_{\text{out},j} \rangle(t) = \langle \hat{b}_{\text{in},j} \rangle(t) + \sqrt{\kappa_j}\langle \hat{a} \rangle(t). \tag{4}$$

To solve these equations, we approximate[54,55]

$$\langle \hat{a}\hat{\sigma}_z \rangle(t) \rightarrow \langle \hat{a} \rangle(t)\langle \hat{\sigma}_z \rangle, \tag{5}$$

where $\langle \hat{\sigma}_z \rangle$ is evaluated at steady state and captures the difference between the population of the excited qubit state and the ground state, accounting for operation at larger temperatures or drive strengths. In our experiments, we operate in the linear regime, $\langle \hat{\sigma}_z \rangle = -1$.

To compute the transmission amplitude, we solve Eqs. (3) and (4) upon Fourier transformation and set $\langle \hat{b}_{\text{in},2} \rangle(t) = 0$. This results in the transmission amplitude

$$\tau(\omega) = -\frac{\langle \hat{b}_{\text{out},2} \rangle(\omega)}{\langle \hat{b}_{\text{in},1} \rangle(\omega)} = \sqrt{\kappa_1\kappa_2}A(\omega), \tag{6}$$

where the minus sign accounts for the phase difference of $\pi$ between the input and the output port ($\lambda/2$ resonator) and

$$A(\omega) = \frac{\gamma + i(\omega_q - \omega)}{[\kappa/2 + i(\omega_0 - \omega)][\gamma + i(\omega_q - \omega)] - g^2\langle \hat{\sigma}_z \rangle}. \tag{7}$$

In the main text, the absolute value squared of this quantity normalized by its maximal value is shown.

The phase of the transmitted signal is given by

$$\varphi(\omega) = -\arctan(\Lambda),$$
$$\Lambda = \frac{-2(\omega_q - \omega)g^2\langle \hat{\sigma}_z \rangle - 2(\omega_0 - \omega)[\gamma^2 + (\omega_q - \omega)^2]}{\kappa[\gamma^2 + (\omega_q - \omega)^2] - 2\gamma g^2 \langle \hat{\sigma}_z \rangle}. \tag{8}$$

As examples, the phase and amplitude of the bare resonance in Coulomb blockade is simultaneously fit in Fig. S1a in the Supplementary and in Fig. S3 in Supplementary the same is done for a linecut of Fig. 4a at 0.25 T.

## Estimation of the photon number

Similarly, we may obtain $\langle \hat{a} \rangle(t)$ by solving Eq. (3). Using $\langle \hat{b}_{\text{in},1} \rangle(t) = \exp(-i\omega_p t)\sqrt{P_{\text{in}}/\omega_p}$, where $P_{\text{in}}$ denotes the power in the input field, we find

$$\langle \hat{a} \rangle(t) = -\sqrt{\frac{\kappa_1 P_{\text{in}}}{\hbar\omega_p}}e^{-i\omega_p t}A(\omega_p). \tag{9}$$

In the low-drive regime we consider here, we estimate the photon number as

$$n = |\langle \hat{a} \rangle|^2 = \frac{\kappa_1 P_{\text{in}}}{\hbar\omega_p}|A(\omega_p)|^2, \tag{10}$$

where we approximate $\kappa_1 \simeq \kappa/2$.

## Effective two-electron Hamiltonian model

We model an effective two-electron Hamiltonian in the presence of spin–orbit interaction and magnetic field. We write the Hamiltonian in the basis of singlet and triplet states $\{|S_{1,1}\rangle, |S_{2,0}\rangle, |T_{1,1}^{\pm,0}\rangle, |T_{2,0}^{\pm,0}\rangle\}$, with the subscripts indicating the charge distribution in the DQD. The Hamiltonian reads

$$\mathcal{H} = \mathcal{H}_0^S + \mathcal{H}_0^T + \mathcal{H}_Z + \mathcal{H}_{\text{so}}, \tag{11}$$

with the spin quantum-number conserving Hamiltonians

$$\mathcal{H}_0^S/\hbar = -\varepsilon|S_{2,0}\rangle\langle S_{2,0}| + t_c^S|S_{1,1}\rangle\langle S_{2,0}| + \text{h.c.}, \tag{12}$$

$$\mathcal{H}_0^T/\hbar = (\Delta_{\text{ST}} - \varepsilon)\sum_{\pm,0}|T_{2,0}^{\pm,0}\rangle\langle T_{2,0}^{\pm,0}|$$
$$+ t_c^T\sum_{\pm,0}|T_{1,1}^{\pm,0}\rangle\langle T_{2,0}^{\pm,0}| + \text{h.c.} \tag{13}$$

Here, $t_c^{S,T}$ are the tunnel rates between the two singlets, and between the two triplet states respectively, and $\Delta_{\text{ST}}$ is the single-dot singlet–triplet splitting that separates the $T_{2,0}$ states from the $S_{2,0}$ states. The Zeeman Hamiltonian is given by

$$\mathcal{H}_Z/\mu_B = B\sum_{\pm}\left(\pm\frac{g_L + g_R}{2}|T_{1,1}^{\pm}\rangle\langle T_{1,1}^{\pm}| \pm g_L|T_{2,0}^{\pm}\rangle\langle T_{2,0}^{\pm}|\right), \tag{14}$$

where $g_L$ ($g_R$) is the Landé $g$-factor of the left (right) dot. Because of the large intrinsic spin–orbit interaction in the NW, we include the spin–orbit Hamiltonian that couples the singlet and triplet states with opposite charge configuration using the spin–orbit tunnel rate $t_{\text{SO}}$ as

$$\mathcal{H}_{\text{SO}}/\hbar = t_{\text{SO}}\left(|T_{1,1}^0\rangle\langle S_{2,0}| + \sum_{\pm}\pm|T_{1,1}^{\pm}\rangle\langle S_{2,0}|\right) + \text{h.c.} \tag{15}$$

## Data availability

The nummerical data used in this study are available in the zenodo database https://doi.org/10.5281/zenodo.7777840.

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

## Acknowledgements

We acknowledge fruitful discussions with Simon Zihlmann, Roy Haller, Andrea Hofmann, Stefano Bosco, Romain Maurand and Antti Ranni, and support in setting-up the experiments by Fabian Oppliger, Roy Haller, Luk Yi Cheung, and Deepankar Sarmah. This research was supported by the following institutions: the Swiss Nanoscience Institute, SNI (J.H.U., A.B., C.S.); the Swiss National Science Foundation through grant 192027 (A.P., C.S.); the NCCR Quantum Science and Technology, NCCR-QSIT (J.H.U., A.P., J.R., C.S.); the NCCR Spin Qubit in Silicon, NCCR-Spin (J.H.U., A.P., A.K., J.R., C.S.); the Eccellenza Professorial Fellowship PCEFP2_194268 (P.P.P.); the European Union's Horizon 2020 research and innovation program, FET-open project AndQC, agreement No. 828948 (C.S.); the European Union's Horizon 2020 research and innovation program, FET-open project TOPSQUAD, agreement No. 847471 (A.P., C.S.); NanoLund (S.L., C.T., K.A.D., V.F.M.); the Knut & Alice Wallenberg Foundation, KAW (K.A.D.); the SNSF through grant 200021_200418 (P.S.); the SERI through grant contract number MB22.00081 (P.S.); European Union's Horizon 2020 research and innovation program: grant agreement No. 787414, ERC-Adv Top-Supra (A.B.).

## Author contributions

J.H.U., A.P., A.K., S.L., J.R., C.T., K.A.D., V.F.M., P.S., A.B., and C.S. developed techniques that were needed to fabricate and measure the device. S.L. grew the nanowire with contributions from C.T. and K.A.D. J.H.U. and A.P. fabricated the device. J.H.U., A.P. and A.K. performed the measurements. P.P.P. devloped the input-output theory allowing for quantitative data analysis. J.H.U. and A.P. analyzed the data with input from all authors. J.H.U. and A.P. wrote the manuscript with contributions from all authors. C.S. supervised the project.

## Competing interests

The authors declare no competing interests.
