## [Peer Review File · Nature Communications]

Reviewer 2:

The authors have done a lot of work, both on the experimental and analysis sides for this resubmission. I appreciate that they have considered my remarks seriously and could answer my concerns. In particular figure 3 now indeed shows that the strong coupling regime has been achieved in their device and the corresponding discussion is now clear and correct. In the new version, I still have a remark though in that the reason for the magnetospectroscopy to be done at a different angle is not explained. Is there any reason?

Although the strong coupling regime is established now, the data show quite a poor quality signal which is not really on par with other platforms, even considering first experimental realization 5 to 6 years ago. More importantly, I am a bit concerned by the very large decoherence rate of the qubit with a lowest value of 116MHz. How spin like is that? Is it expected for singlet-triplet qubit in this material to have so high decoherence intrinsically or is there any degradation in the cQED architecture as is observed for Si spin qubits in cavity? The most important concern comes from the acknowledgment by the authors that due to such a large decoherence, they cannot perform two-tone spectroscopy, and even less do qubit manipulation. I therefore cannot see this architecture as a possible qubit platform if the qubit cannot be resolved spectroscopically nor manipulated. In addition, the authors have now measured four devices in total where I understand they could not do that. Changing the magnetic field angle appears to degrade further the system coherence and would therefore not be a way to improve the system. Given the results of this paper and the discussion with the authors, it is rather difficult at this stage to imagine that this platform could ever produce useful qubits. I believe it is worth publishing though, but maybe not in Nature Physics. Nature communications would be better suited in my opinion.

Reviewer 2: The authors have done a lot of work, both on the experimental and analysis sides for this resubmission. I appreciate that they have considered my remarks seriously and could answer my concerns. In particular figure 3 now indeed shows that the strong coupling regime has been achieved in their device and the corresponding discussion is now clear and correct.

We thank Reviewer 2 for appreciating our work and, most importantly, we are pleased to read that Reviewer 2 now agrees that we have established strong coupling, which is the main claim of our manuscript.

In the new version, I still have a remark though in that the reason for the magnetospectroscopy to be done at a different angle is not explained. Is there any reason?

Changing the in-plane angle of the magnetic field has a vast impact the qubit transition frequency, coupling strength and coherence. This effect is expected in our theoretical framework, because of an altering projection of the spin-orbit vector onto the applied magnetic field as the field rotates. The impact

of the field rotation on the qubit parameters, e.g. decoherence rate and transition frequency, go beyond the scope of our manuscript and are explored in a paper which we are currently preparing. To clarify this, we added the following statement in the corresponding section of the manuscript: 'The dependence of the qubit parameters on the angle of the in-plane magnetic field is beyond the scope of this manuscript and will be investigated in future studies.'

Although the strong coupling regime is established now, the data show quite a poor quality signal which is not really on par with other platforms, even considering first experimental realization 5 to 6 years ago. More importantly, I am a bit concerned by the very large decoherence rate of the qubit with a lowest value of 116MHz. How spin like is that? Is it expected for singlet-triplet qubit in this material to have so high decoherence intrinsically or is there any degradation in the cQED architecture as is observed for Si spin qubits in cavity?

In fact, the decoherence rate reported here is similar to the ones of previously reported single electron spin-qubits in semiconductor nanowires [Pettersson, Karl D., et al. *Nature* 490.7420 (2012); Froning, Florian NM, et al. *Nature Nanotechnology* 16.3 (2021)]. But, advanced pulsing protocols such as dynamical decoupling promise smaller decoherence rates [Nadj-Perge, S., et al. *Nature* 468.7327 (2010); Barthel, C., et al. *Physical review letters* 105.26 (2010)] in the future.

The origin of decoherence by itself is an exciting field of study which goes beyond the scope of the here presented manuscript. We will investigate the decoherence mechanisms in detail in the above mentioned manuscript which is currently in preparation.

The most important concern comes from the acknowledgment by the authors that due to such a large decoherence, they cannot perform two-tone spectroscopy, and even less do qubit manipulation. I therefore cannot see this architecture as a possible qubit platform if the qubit cannot be resolved spectroscopically nor manipulated. In addition, the authors have now measured four devices in total where I understand they could not do that.

Strong spin-photon coupling is a critical step in establishing a new qubit platform. We did not aim to demonstrate everything at once for this platform, as it took many Nature papers for others. But this platform is fundamentally

different from all previous ones and might be a game changer since many issues in scaling up come from getting each qubit tuned correctly. The significant reproducibility and growth-determined qubit parameters underscore the excellence of our platform for scalability. Our results are very fundamental. Therefore, they establish a new platform and form a proof-of-principle, which will broadly impact other material platforms.

Changing the magnetic field angle appears to degrade further the system coherence and would therefore not be a way to improve the system.

Reviewer 2 assumes here that changing the magnetic field angle will result in worse coherence, but the opposite is true. Changing the magnetic field angle can result in superior coherence, with the opportunity to be further optimised in future devices. As mentioned above, we discuss these results in a manuscript that is currently in preparation.

Given the results of this paper and the discussion with the authors, it is rather difficult at this stage to imagine that this platform could ever produce useful qubits.

From interactions with other researchers, we are certain that our work is very exciting for the spin-qubit community, as we are aware of several research groups who have re-aligned their research to perform follow-up experiments.

I believe it is worth publishing though, but maybe not in Nature Physics. Nature communications would be better suited in my opinion.

We are happy to read that Reviewer 2 suggests publication of our work in Nature Communications.